# INTERPRETING AND IMPROVING DIFFUSION MODELS USING THE EUCLIDEAN DISTANCE FUNCTION

## ABSTRACT

Denoising is intuitively related to projection. Indeed, under the manifold hypothesis, adding random noise is approximately equivalent to orthogonal perturbation. Hence, learning to denoise is approximately learning to project. In this paper, we use this observation to reinterpret denoising diffusion models as approximate gradient descent applied to the Euclidean distance function. We then provide straight-forward convergence analysis of the DDIM sampler under simple assumptions on the projection-error of the denoiser. Finally, we propose a new sampler based on two simple modifications to DDIM using insights from our theoretical results. In as few as 5-10 function evaluations, our sampler achieves state-of-the-art FID scores on pretrained CIFAR-10 and CelebA models and can generate high quality samples on latent diffusion models.

## 1 INTRODUCTION

Diffusion models achieve state-of-the-art quality on many image generation tasks (Ramesh et al., 2022; Rombach et al., 2022; Saharia et al., 2022). They are also successful in text-to-3D generation (Poole et al., 2022) and novel view synthesis (Liu et al., 2023). Outside the image domain, they have been used for robot path-planning (Chi et al., 2023), prompt-guided human animation (Tevet et al., 2022), and text-to-audio generation (Kong et al., 2020).

Diffusion models are presented as the reversal of a stochastic process that corrupts clean data with increasing levels of random noise (Sohl-Dickstein et al., 2015; Ho et al., 2020). This reverse process can also be interpreted as likelihood maximization of a noise-perturbed data-distribution using learned gradients (called *score functions*) (Song & Ermon, 2019; Song et al., 2020b). While these interpretations are inherently probabilistic, samplers widely used in practice (e.g. Song et al. (2020a)) are often deterministic. In this paper, we tackle this divide and provide a deterministic framework for reasoning about, improving and potentially discovering new applications of diffusion models.

For our first contribution, we reinterpret diffusion models as *projection* onto the *support* of the training-set distribution, discarding the underlying measure. This deterministic interpretation is based on an approximate correspondence between denoising and projection (noted in Chung et al. (2022); Rick Chang et al. (2017)) that we make rigorous in Section 3, assuming the manifold hypothesis. We then reinterpret sampling as approximate gradient descent on the Euclidean distance-function and perform convergence analysis under a simple error model relating denoising and projection (Section 4). This analysis also provides rigorous justification for log-linear noise schedules. Finally, we leverage properties of the distance function to design a high-order sampler that aggregates previous denoiser outputs to reduce error (Section 5).

We conclude with computational evaluation of our sampler (Section 6) that demonstrates state-of-the-art FID scores on pretrained CIFAR-10 and CelebA datasets and comparable results to the best samplers for high-resolution latent models such as Stable Diffusion (Rombach et al., 2022) (Figure 1). Section 7 provides novel interpretations of existing techniques under the framework of distance functions and outlines directions for future research.

| **Ours** | **UniPC** | **DPM++** | **PNDM** | **DDIM** |
|---|---|---|---|---|
| | (Zhao et al., 2023) | (Lu et al., 2022b) | (Liu et al., 2022) | (Song et al., 2020a) |
| **FID 13.77** | 15.59 | 15.43 | 19.43 | 14.06 |

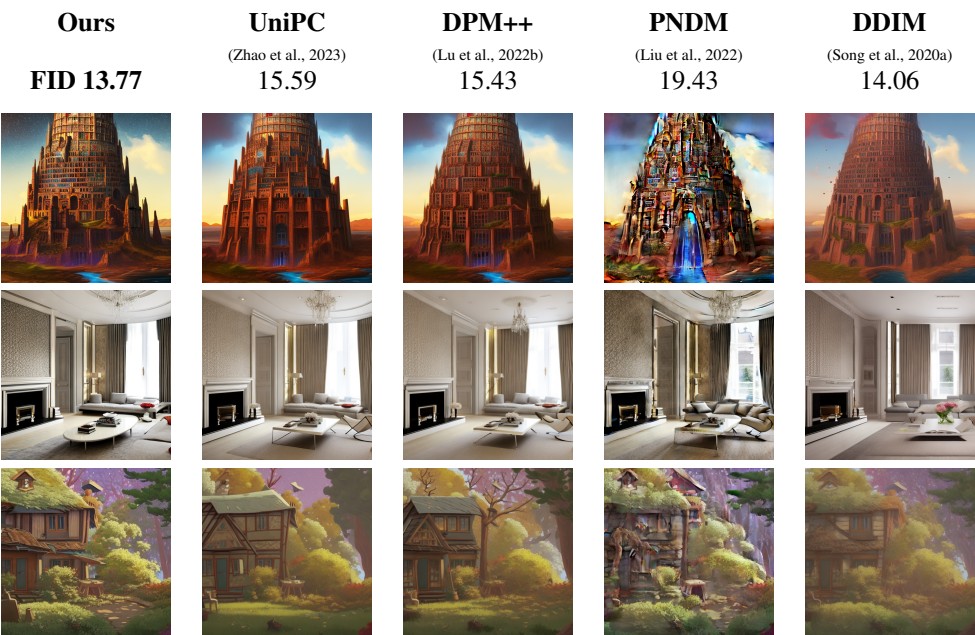

Figure 1: Outputs of our sampler on text-to-image Stable Diffusion compared to other commonly used samplers, when limited to $N = 10$ function evaluations. We also report FID scores for text-to-image generation on MS-COCO 30K.

## 2 BACKGROUND

Denoising diffusion models (along with all other generative models) treat datasets as samples from a probability distribution supported on a subset $\mathcal{K}$ of $\mathbb{R}^n$. They are used to *generate* new points in $\mathcal{K}$ outside the training set. We overview their basic features. We then state properties of the Euclidean distance function $\text{dist}_{\mathcal{K}}(x)$ that are key to our contributions.

### 2.1 DENOISING DIFFUSION MODELS

**Denoisers** Denoising diffusion models are trained to estimate a *noise vector* $\epsilon \in \mathbb{R}^n$ from a given noise level $\sigma > 0$ and noisy input $x_\sigma \in \mathbb{R}^n$ such that $x_\sigma = x_0 + \sigma\epsilon$ approximately holds for some $x_0$ in the data manifold $\mathcal{K}$. The learned function, denoted $\epsilon_\theta : \mathbb{R}^n \times \mathbb{R}_+ \to \mathbb{R}^n$, is called a *denoiser*. The trainable parameters, denoted jointly by $\theta \in \mathbb{R}^m$, are found by (approximately) minimizing

$$L(\theta) := \mathbf{E}_{x,\sigma,\epsilon} \|\epsilon_\theta(x_0 + \sigma\epsilon, \sigma) - \epsilon\|^2 \tag{1}$$

when $x_0$ is drawn from the training-set distribution, $\sigma$ is drawn uniformly from a finite set of positive numbers, and $\epsilon$ is drawn from a Gaussian distribution $\mathcal{N}(0, I)$. In practice, training is done by applying stochastic gradient descent to $L(\theta)$ using randomly sampled $(x, \epsilon, \sigma)$.

Throughout, we let $\{\sigma_t\}_{t=0}^N$ denote the monotonically increasing $\sigma$ *schedule*. For simplicity of notation we use $\epsilon_\theta(\cdot, \sigma_t)$ and $\epsilon_\theta(\cdot, t)$ interchangeably based on context. The sequence of $\sigma_t$ is the basis of *sampling algorithms* we overview next.

**Sampling** Given noisy $x_\sigma$ and noise level $\sigma$, the denoiser $\epsilon_\theta(x_\sigma, \sigma)$ induces an estimate of $\hat{x}_0 \approx x_0$ via

$$\hat{x}_0(x_\sigma, \sigma) := x_\sigma - \sigma\epsilon_\theta(x_\sigma, \sigma). \tag{2}$$

Aiming to improve accuracy, sampling algorithms construct a sequence $\hat{x}_0^t := \hat{x}_0(x_t, \sigma_t)$ of estimates that in turn arises from a sequence of points $x_t$ initialized at a given $x_N$. The most basic samplers recursively construct $x_{t-1}$ from $x_t$ and $\epsilon_\theta(x_t, \sigma_t)$. For instance, the DDPM (Ho et al., 2020) sampler uses the recursion

$$x_{t-1} = x_t + (\sigma_{t'} - \sigma_t)\epsilon_\theta(x_t, \sigma_t) + \eta w_t, \tag{3}$$

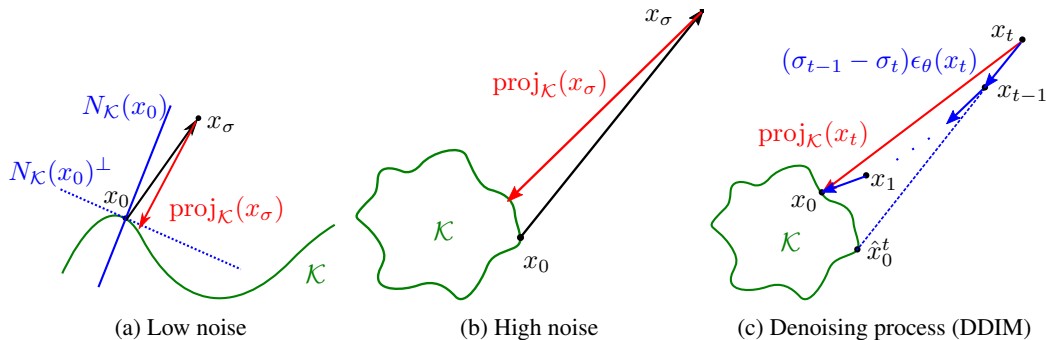

Figure 2: Denoising approximates projection: When $\sigma$ is small (2a), most of the added noise lies in $N_{\mathcal{K}}(x_0)$ with high probability under the manifold hypothesis. When $\sigma$ is large (2b), both denoising and projection point in the same direction towards $\mathcal{K}$. We interpret the denoising process (2c) as finding $x_0 \in \mathcal{K}$ by iteratively estimating $\text{proj}_{\mathcal{K}}(x_t)$ with $\epsilon_\theta(x_t)$.

where $w_t \sim \mathcal{N}(0, I)$, $\sigma_{t'} = \sigma_{t-1}^2 / \sigma_t$ and $\eta = \sqrt{\sigma_{t-1}^2 - \sigma_{t'}^2}$. (Note that by definition $\sigma_{t'} < \sigma_{t-1} < \sigma_t$, as $\sigma_{t-1}$ is the geometric mean of $\sigma_{t'}$ and $\sigma_t$.) The DDIM (Song et al., 2020a) sampler, on the other hand, uses the recursion

$$x_{t-1} = x_t + (\sigma_{t-1} - \sigma_t)\epsilon_\theta(x_t, \sigma_t). \tag{4}$$

DDPM is randomized given the samples $w_t$ whereas DDIM is deterministic. See Figure 2c for an illustration of this denoising process. Note that these samplers were originally presented in variables $z_t$ satisfying $z_t = \sqrt{\alpha_t} x_t$, where $\alpha_t$ satisfies $\sigma_t^2 = \frac{1-\alpha_t}{\alpha_t}$. We prove equivalence of the original definitions to (3) and (4) in Appendix A and note that the change-of-variables from $z_t$ to $x_t$ previously appears in Song et al. (2020b); Karras et al. (2022); Song et al. (2020a).

## 2.2 Distance, Projection, and Reach

The *distance function* of a set $\mathcal{K} \subseteq \mathbb{R}^n$, denoted $\text{dist}_{\mathcal{K}} : \mathbb{R}^n \to \mathbb{R}$, is defined via

$$\text{dist}_{\mathcal{K}}(x) := \inf\{\|x - x_0\| : x_0 \in \mathcal{K}\}. \tag{5}$$

The *projection* of $x \in \mathbb{R}^n$, denoted $\text{proj}_{\mathcal{K}}(x)$, is the set of points that attain this distance, i.e.,

$$\text{proj}_{\mathcal{K}}(x) := \{x_0 \in \mathcal{K} : \text{dist}_{\mathcal{K}}(x) = \|x - x_0\|\}. \tag{6}$$

When $\text{proj}_{\mathcal{K}}(x)$ is a singleton, i.e., when $\text{proj}_{\mathcal{K}}(x) = \{x_0\}$, we abuse notation and let $\text{proj}_{\mathcal{K}}(x)$ denote $x_0$. We collect useful facts below.

**Proposition 2.1** (page 283, Theorem 3.3 of Delfour & Zolésio (2011)). *Suppose $\mathcal{K} \subseteq \mathbb{R}^n$ is closed and $x \notin \mathcal{K}$. If $\text{proj}_{\mathcal{K}}(x)$ is a singleton, then the following statements hold:*

- *The gradient $\nabla\text{dist}_{\mathcal{K}}(x)$ exists and satisfies $\nabla\text{dist}_{\mathcal{K}}(x) = \text{dist}_{\mathcal{K}}(x)^{-1}(x - \text{proj}_{\mathcal{K}}(x))$. Further, $\|\nabla\text{dist}_{\mathcal{K}}(x)\| = 1$.*

- *The gradient of $f(x) := \frac{1}{2}\text{dist}_{\mathcal{K}}(x)^2$ satisfies $\nabla f(x) = \text{dist}_{\mathcal{K}}(x)\nabla\text{dist}_{\mathcal{K}}(x)$. Equivalently, $\nabla f(x) = x - \text{proj}_{\mathcal{K}}(x)$.*

*Further, $\text{proj}_{\mathcal{K}}(x)$ is a singleton for almost all $x \in \mathbb{R}^n$ (under the Lebesgue measure).*

Suppose that $x_\sigma = x_0 + \sigma\epsilon$ for $\epsilon \sim \mathcal{N}(0, I)$. The unit vector $\nabla\text{dist}_{\mathcal{K}}(x_\sigma)$ is intuitively related to $\epsilon$ whereas $\text{dist}_{\mathcal{K}}(x_\sigma)$ is intuitively related to $\sigma$. The next section establishes conditions when these quantities are approximately equal up to scaling by $\sqrt{n}$. We will state results using the *reach* of $\mathcal{K}$, defined as the largest $\tau > 0$ such that $\text{proj}_{\mathcal{K}}(x)$ is unique when $\text{dist}_{\mathcal{K}}(x) < \tau$. Local Lipschitz continuity of $\text{proj}_{\mathcal{K}}(x)$ can also be established using $\text{reach}(\mathcal{K})$; see Appendix B.

## 3 MANIFOLD HYPOTHESIS AND EQUIVALENCE BETWEEN DENOISING AND PROJECTION

The *manifold hypothesis* (Bengio et al., 2013; Fefferman et al., 2016; Pope et al., 2021) asserts that "real-world" datasets are (approximately) contained in low-dimensional manifolds of $\mathbb{R}^n$. We next show that denoising is approximately equivalent to projection under a version of this hypothesis. Specifically, we suppose that $\mathcal{K}$ is a manifold of dimension $d$ with $d \ll n$. Given $x_\sigma = x_0 + \sigma\epsilon$, we then show that the denoiser $\epsilon_\theta(x_\sigma, \sigma)$ approximates $\mathrm{proj}_{\mathcal{K}}(x_\sigma)$ with error that decreases with $d$. This is because most of the noise lie in the normal space $N_{\mathcal{K}}(x_0) \subseteq \mathbb{R}^n$, a subspace of dimension $n - d$, and perturbations in $N_{\mathcal{K}}(x_0)$ do not change the projection (see Figure 2a for an illustration):

**Lemma 3.1** (Theorem 4.8(12) in Federer (1959)). *Given $x_0 \in \mathcal{K}$ and $w \in N_{\mathcal{K}}(x)$, if $\|w\| <$ reach$(\mathcal{K})$, then $\mathrm{proj}_{\mathcal{K}}(x_0 + w) = x_0$.*

Decomposing $\epsilon$ using $N_{\mathcal{K}}(x_0) \oplus N_{\mathcal{K}}(x_0)^\perp$ and combining Lemma 3.1 with Gaussian concentration inequalities (Vershynin, 2018, Chapter 3) provides a *backward error* bound on the approximation $\mathrm{proj}_{\mathcal{K}}(x_\sigma) \approx x_0$. Invoking Lipschitz continuity of $\mathrm{proj}_{\mathcal{K}}(x_\sigma)$ when $\mathrm{dist}_{\mathcal{K}}(x_\sigma) < \mathrm{reach}(\mathcal{K})$ provides a *forward error* bound. We state these bounds informally here, deferring precise statements and proofs to Appendix B.

**Theorem 3.1** (Denoising vs Projection (informal)). *Fix $\sigma > 0$ and suppose that $\mathrm{reach}(\mathcal{K}) \gtrsim \sigma\sqrt{n}$. Given $x_0 \in \mathcal{K}$ and $\epsilon \sim \mathcal{N}(0, I)$, let $x_\sigma = x_0 + \sigma\epsilon$. With high probability, we have:*

- *(Backward error) $x_0 = \mathrm{proj}_{\mathcal{K}}(x_\sigma + \delta)$ for some $\delta \in \mathbb{R}^n$ satisfying $\|\delta\| \leq \sigma\sqrt{d}$.*

- *(Forward error) $\|\mathrm{proj}_{\mathcal{K}}(x_\sigma) - x_0\| \lesssim \sigma\sqrt{d}$.*

The second statement illustrates that perfect denoising of $x_\sigma$ approximates $\mathrm{proj}_{\mathcal{K}}(x_\sigma)$. From Proposition 2.1, it is therefore natural to bound the approximation error of $\epsilon_\theta(x_\sigma, \sigma) \approx \sqrt{n}\nabla\mathrm{dist}_{\mathcal{K}}(x_\sigma)$ and $\sqrt{n}\sigma_t \approx \mathrm{dist}_{\mathcal{K}}(x_\sigma)$, which together induces the approximation $\mathrm{proj}_{\mathcal{K}}(x_\sigma) \approx x_\sigma - \sigma\epsilon_\theta(x_\sigma, \sigma)$. Diffusion models often add large levels of noise to $x_0$ when training the denoiser. In this case, even though the reach assumption may be violated, because the noise level is much larger than the diameter of the data manifold, denoising is still approximately projection in terms of relative error (see Figure 2b). Section 4 incorporates precise versions of these approximations into sampling algorithms. We note that De Bortoli (2022) also analyze diffusion under the manifold hypothesis.

## 4 EQUIVALENCE BETWEEN SAMPLING AND GRADIENT DESCENT

Section 3 establishes that denoising approximates projection. We now study DDIM under two different approximation models. Letting $f(x) := \frac{1}{2}\mathrm{dist}_{\mathcal{K}}(x)^2$ and noting that $\nabla f(x) = x - \mathrm{proj}_{\mathcal{K}}(x)$ when $\nabla f(x)$ exists, we state these models as the following assumptions.

**Assumption 1** (Exact projection). *If $(x, t)$ satisfies $\mathrm{dist}_{\mathcal{K}}(x) = \sqrt{n}\sigma_t$ and $\nabla f(x)$ exists, then $\sigma_t\epsilon_\theta(x, \sigma_t) = \nabla f(x)$.*

**Assumption 2** (Projection with relative error). *There exists $\nu \geq 1$ and $\eta \geq 0$ such that if $\frac{1}{\nu}\mathrm{dist}_{\mathcal{K}}(x) \leq \sqrt{n}\sigma_t \leq \nu\mathrm{dist}_{\mathcal{K}}(x)$ and $\nabla f(x)$ exists, then $\|\sigma_t\epsilon_\theta(x, t) - \nabla f(x)\| \leq \eta\mathrm{dist}_{\mathcal{K}}(x)$.*

Assumption 1 states that denoising is precisely projection, which is unrealistic but gives intuition for the approximations we will later make. Assumption 2 weakens exactness to a relative-error assumption on $\nabla f(x)$ given that $\|\nabla f(x)\| = \mathrm{dist}_{\mathcal{K}}(x)$. We first show that DDIM is precisely gradient descent with step-size determined by $\sigma_t$ under Assumption 1. We then interpret DDIM as gradient descent with relative-error under Assumption 2 and provide simple convergence analysis. Proofs are postponed to Appendix C. Our experiments in Appendix E on image datasets show that the learned denoiser outputs $\epsilon_\theta(x_t, t)$ approximately point in the same direction for all $t$ in the denoising process (i.e. has high cosine similarity) and has approximately unit norm, validating our assumption that denoising is approximately projection for all noise levels.

### 4.1 EXACT PROJECTION AND GRADIENT DESCENT

We use the following lemma for gradient descent applied to the squared-distance function $f(x)$.

**Lemma 4.1.** *Fix $x \in \mathbb{R}^n$ and suppose that $\nabla f(x)$ exists. For step-size $0 < \beta \leq 1$ consider the gradient descent iteration applied to $f(x)$:*

$$x_+ := x - \beta \nabla f(x)$$

*Then,* $\mathrm{dist}_{\mathcal{K}}(x_+) = (1 - \beta) \, \mathrm{dist}_{\mathcal{K}}(x) < \mathrm{dist}_{\mathcal{K}}(x)$.

We can now characterize DDIM as follows.

**Theorem 4.1.** *Let $x_N, x_{N-1}, \ldots, x_0$ denote a sequence (18) generated by DDIM and suppose that Assumption 1 holds. Further suppose that the gradient of $f(x) := \frac{1}{2}\mathrm{dist}_{\mathcal{K}}(x)^2$ exists for all $x_t$. Then the following statements hold:*

- *$x_t$ equals the sequence generated by gradient descent with step-size $\beta_t := 1 - \sigma_{t-1}/\sigma_t$,*

$$x_{t-1} = x_t - \beta_t \nabla f(x_t).$$

- *$\mathrm{dist}_{\mathcal{K}}(x_t) = \sqrt{n}\sigma_t$ for all $t$.*

We remark that $\mathrm{dist}_{\mathcal{K}}(x_t) < \mathrm{reach}(\mathcal{K})$ will *guarantee* the existence of $\nabla f(x_t)$ for each $t$, but the existance of $\nabla f(x_t)$ is a weak assumption as it will be satisfied by almost all $x \in \mathbb{R}^n$.

## 4.2 APPROXIMATE PROJECTION AND GRADIENT DESCENT WITH ERROR

Our relative-error model (Assumption 2) supposes that $\frac{1}{\nu}\mathrm{dist}_{\mathcal{K}}(x) \leq \sqrt{n}\sigma_t \leq \nu\mathrm{dist}_{\mathcal{K}}(x)$. To ensure this condition holds at each DDIM iteration, we need to lower and upper bound distance. For this, we use the following two lemmas.

**Lemma 4.2.** *The distance function $\mathrm{dist}_{\mathcal{K}} : \mathbb{R}^n \to \mathbb{R}$ for $\mathcal{K} \subseteq \mathbb{R}^n$ satisfies*

$$\mathrm{dist}_{\mathcal{K}}(u) - \|u - v\| \leq \mathrm{dist}_{\mathcal{K}}(v) \leq \mathrm{dist}_{\mathcal{K}}(u) + \|u - v\|$$

*for all $u, v \in \mathbb{R}^n$.*

**Lemma 4.3.** *For $\mathcal{K} \subseteq \mathbb{R}^n$, let $f(x) := \frac{1}{2}\mathrm{dist}_{\mathcal{K}}(x)^2$. If $x_{t-1} = x_t - \beta_t(\nabla f(x_t) + e_t)$ for $e_t$ satisfying $\|e_t\| \leq \eta\mathrm{dist}_{\mathcal{K}}(x_t)$ and $0 \leq \beta_t \leq 1$, then*

$$\mathrm{dist}_{\mathcal{K}}(x_N) \prod_{i=t}^{N}(1 - \beta_i(\eta + 1)) \leq \mathrm{dist}_{\mathcal{K}}(x_{t-1}) \leq \mathrm{dist}_{\mathcal{K}}(x_N) \prod_{i=t}^{N}(1 + \beta_i(\eta - 1).) \tag{7}$$

From Lemma 4.3, the following condition ensures that $\frac{1}{\nu}\mathrm{dist}_{\mathcal{K}}(x) \leq \sqrt{n}\sigma_t \leq \nu\mathrm{dist}_{\mathcal{K}}(x)$ holds at each DDIM iteration, leveraging the DDIM property that $\sigma_{t-1} = (1 - \beta_t)\sigma_t$.

**Definition 4.1.** *We say that parameters $\{\sigma_t\}_{t=0}^{N}$ are $(\eta, \nu)$-admissible if, for all $t \in \{1, \ldots, N\}$,*

$$\frac{1}{\nu}\prod_{i=t}^{N}(1 + \beta_i(\eta - 1)) \leq \prod_{i=t}^{N}(1 - \beta_i) \leq \nu\prod_{i=t}^{N}(1 - \beta_i(\eta + 1)), \tag{8}$$

*where $\beta_t := 1 - \sigma_{t-1}/\sigma_t$.*

We now give error bounds for DDIM under the assumption that the noise levels $\sigma_t$ are admissible. We then study admissible sequences for which $\sigma_{t-1}/\sigma_t$ is constant, which in turn implies that the DDIM step-size $\beta_t$ is fixed (Theorem 4.1).

### 4.2.1 ERROR BOUNDS

Our main result under the relative-error model follows.

**Theorem 4.2** (DDIM with relative error). *Let Assumption 2 hold and suppose $\{\sigma_t\}_{t=0}^{N}$ is $(\eta, \nu)$-admissible for $0 \leq \eta < 1$ and $\nu \geq 1$. Let $x_t$ denote the sequence generated by DDIM and suppose that the gradient of $f(x) := \frac{1}{2}\mathrm{dist}_{\mathcal{K}}(x)^2$ exists for all $x_t$. The following statements hold.*

- *$x_t$ is generated by approximate gradient descent iterations of the form (7) in Lemma 4.3 with $\beta_t = 1 - \sigma_{t-1}/\sigma_t$.*

- *$\frac{1}{\nu}\mathrm{dist}_{\mathcal{K}}(x_t) \leq \sqrt{n}\sigma_t \leq \nu\mathrm{dist}_{\mathcal{K}}(x_t)$ for all $t$.*

- *$\mathrm{dist}_{\mathcal{K}}(x_N) \prod_{i=t}^{N}(1 - \beta_i(\eta + 1)) \leq \mathrm{dist}_{\mathcal{K}}(x_{t-1}) \leq \mathrm{dist}_{\mathcal{K}}(x_N) \prod_{i=t}^{N}(1 + \beta_i(\eta - 1))$.*

| **Algorithm 1** DDIM sampler (Song et al., 2020a) | **Algorithm 2** Our gradient estimation sampler |
|---|---|
| **Require:** $(\sigma_N, \dots, \sigma_0)$, $x_N \sim \mathcal{N}(0, I)$, $\epsilon_\theta$ | **Require:** $(\sigma_N, \dots, \sigma_0)$, $x_N \sim \mathcal{N}(0, I)$, $\epsilon_\theta$ |
| **Ensure:** Compute $x_0$ with $N$ evaluations of $\epsilon_\theta$ | **Ensure:** Compute $x_0$ with $N$ evaluations of $\epsilon_\theta$ |
|    **for** $t = N, \dots, 1$ **do** |    $x_{N-1} \leftarrow x_N + (\sigma_{N-1} - \sigma_N)\epsilon_\theta(x_N, \sigma_N)$ |
|       $x_{t-1} \leftarrow x_t + (\sigma_{t-1} - \sigma_t)\epsilon_\theta(x_t, \sigma_t)$ |    **for** $t = N-1, \dots, 1$ **do** |
|    **return** $x_0$ |       $\bar{\epsilon}_t \leftarrow 2\epsilon_\theta(x_t, \sigma_t) - \epsilon_\theta(x_{t+1}, \sigma_{t+1})$ |
| |       $x_{t-1} \leftarrow x_t + (\sigma_{t-1} - \sigma_t)\bar{\epsilon}_t$ |
| |    **return** $x_0$ |

### 4.2.2 Admissible Log-Linear Schedules for DDIM

We next characterize admissible $\sigma_t$ of the form $\sigma_{t-1} = (1 - \beta)\sigma_t$ where $\beta$ denotes a *constant* step-size. This illustrates that admissible $\sigma_t$-sequences not only *exist*, they can also be explicitly constructed from $(\eta, \nu)$.

**Theorem 4.3.** *Fix* $\beta \in \mathbb{R}$ *satisfying* $0 \le \beta < 1$ *and suppose that* $\sigma_{t-1} = (1 - \beta)\sigma_t$. *Then* $\sigma_t$ *is* $(\eta, \nu)$*-admissible if and only if* $\beta \le \beta_{*,N}$ *where* $\beta_{*,N} := \frac{c}{\eta + c}$ *for* $c := 1 - \nu^{-1/N}$.

Suppose we fix $(\eta, \nu)$ and choose, for a given $N$, the step-size $\beta_{*,N}$. It is natural to ask how the error bounds of Theorem 4.2 change as $N$ increases. The following establishes the limiting behavior of the *final* output $(\sigma_0, x_0)$ of DDIM.

**Theorem 4.4.** *Let* $x_N, \dots, x_1, x_0$ *denote the sequence generated by DDIM with* $\sigma_t$ *satisfying* $\sigma_{t-1} = (1 - \beta_{*,N})\sigma_t$ *for* $\nu \ge 1$ *and* $\eta > 0$. *The following statements hold*

- $\lim_{N \to \infty} \sigma_0 / \sigma_N = \lim_{N \to \infty} (1 - \beta_{*,N})^N = (1/\nu)^{1/\eta}$.

- $\lim_{N \to \infty} \text{dist}_{\mathcal{K}}(x_0) / \text{dist}_{\mathcal{K}}(x_N) \le \lim_{N \to \infty} (1 + (\eta - 1)\beta_{*,N})^N = (1/\nu)^{\frac{1-\eta}{\eta}}$.

This theorem illustrates that final error, while bounded, need not converge to zero under our error model. This motivates heuristically updating the step-size from $\beta_{*,N}$ to a full step ($\beta = 1$) during the final DDIM iteration. We adopt this approach in our experiments (Section 6).

## 5 Improving Deterministic Sampling Algorithms via Gradient Estimation

Section 3 establishes that $\epsilon_\theta(x, \sigma) \approx \sqrt{n}\nabla\text{dist}_{\mathcal{K}}(x)$ when $\text{dist}_{\mathcal{K}}(x) \approx \sqrt{n}\sigma$. We next exploit an invariant property of $\nabla\text{dist}_{\mathcal{K}}(x)$ to reduce the prediction error of $\epsilon_\theta$ via *gradient estimation*.

The gradient $\nabla\text{dist}_{\mathcal{K}}(x)$ is *invariant* along line segments between a point $x$ and its projection $\text{proj}_{\mathcal{K}}(x)$, i.e., letting $\hat{x} = \text{proj}_{\mathcal{K}}(x)$,

$$\nabla\text{dist}_{\mathcal{K}}(\theta x + (1 - \theta)\hat{x}) = \nabla\text{dist}_{\mathcal{K}}(x) \qquad \forall \theta \in (0, 1]. \tag{9}$$

Hence, $\epsilon_\theta(x, \sigma)$ should be (approximately) constant on this line-segment under our assumption that $\epsilon_\theta(x, \sigma) \approx \sqrt{n}\nabla\text{dist}_{\mathcal{K}}(x)$ when $\text{dist}_{\mathcal{K}}(x) \approx \sqrt{n}\sigma$. Precisely, for $x_1$ and $x_2$ on this line-segment, we should have

$$\epsilon_\theta(x_1, \sigma_{t_1}) \approx \epsilon_\theta(x_2, \sigma_{t_2}) \tag{10}$$

if $t_i$ satisfies $\text{dist}_{\mathcal{K}}(x_i) \approx \sqrt{n}\sigma_{t_i}$. This property suggests combining previous denoiser outputs $\{\epsilon_\theta(x_i, \sigma_i)\}_{i=t+1}^N$ to estimate $\epsilon_t = \sqrt{n}\nabla\text{dist}_{\mathcal{K}}(x_t)$. We next propose a practical *second-order method* [1] for this estimation that combines the current denoiser output with the previous. Recently introduced *consistency models* (Song et al., 2023) penalize violation of (10) during *training*. Interpreting denoiser output as $\nabla\text{dist}_{\mathcal{K}}(x)$ and invoking (9) offers an alternative justification for these models.

---

[1]This method is second-order in the sense that the update step uses previous values of $\epsilon_\theta$, and should not be confused with second-order derivatives.

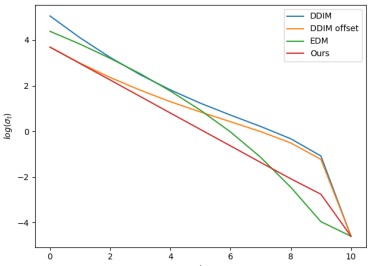

| Schedule | CIFAR-10 | CelebA |
|---|---|---|
| DDIM | 16.86 | 18.08 |
| DDIM Offset | 14.18 | 15.38 |
| EDM | 20.85 | 16.72 |
| Ours | **13.25** | **13.55** |

Figure 3: Plot of different choices of $\log(\sigma_t)$ for $N = 10$.

Table 1: FID scores of the DDIM sampler (Algorithm 1) with different $\sigma_t$ schedules on the CIFAR-10 model for $N = 10$ steps.

Let $e_t(\epsilon) = \epsilon - \epsilon_\theta(x_t, \sigma_t)$ be the error of $\epsilon_\theta(x_t, \sigma_t)$ when predicting $\epsilon$. To estimate $\epsilon$ from $\epsilon_\theta(x_t, \sigma_t)$, we minimize the norm of this error concatenated over two time-steps. Precisely, letting $y_t(\epsilon) = (e_t(\epsilon), e_{t+1}(\epsilon))$, we compute

$$\bar{\epsilon}_t := \arg\min_\epsilon \|y_t(\epsilon)\|_W^2 , \tag{11}$$

where $W$ is a specified positive-definite weighting matrix. In Appendix D we show that this error model results in the update rule

$$\bar{\epsilon}_t = \gamma\epsilon_\theta(x_t, \sigma_t) + (1 - \gamma)\epsilon_\theta(x_{t+1}, \sigma_{t+1}), \tag{12}$$

where we can search over $W$ by searching over $\gamma$.

## 6 EXPERIMENTS

We evaluate modifications of DDIM (Algorithm 1) that leverage insights from Section 5 and Section 4.2.2. Following Section 5 we modify DDIM to use a second-order update that corrects for error in the denoiser output (Algorithm 2). Specifically, we use the Equation (12) update with $\gamma = 2$, which is empirically tuned (see Appendix E). A comparison of this update with DDIM is visualized in Figure 4. Following Section 4.2.2, we select a noise schedule $(\sigma_N, \ldots, \sigma_0)$ that decreases at a log-linear (geometric) rate. The specific rate is determined by an initial and target noise level. Our $\sigma_t$ schedule is illustrated in Figure 3, along with other commonly used schedules. We note that

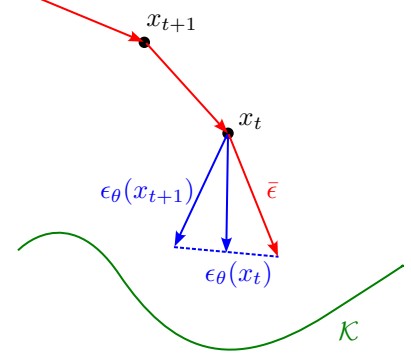

Figure 4: Illustration of our choice of $\bar{\epsilon}_t$

log-linear schedules have been previously proposed for SDE-samplers (Song et al., 2020b); to our knowledge we are the first to propose and analyze their use for DDIM[2]. All the experiments were run on a single Nvidia RTX 4090 GPU.

### 6.1 EVALUATION OF NOISE SCHEDULE

In Figure 3 we plot our schedule (with our choices of $\sigma_t$ detailed in Appendix F) with three other commonly used schedules on a log scale. The first is the evenly spaced subsampling of the training noise levels used by DDIM. The second "DDIM Offset" uses the same even spacing but starts at a smaller $\sigma_N$, the same as that in our schedule. This type of schedule is typically used for guided image generation such as SDEdit (Meng et al., 2021). The third "EDM" is the schedule used in Karras et al. (2022, Eq. 5), with $\sigma_{\max} = 80, \sigma_{\min} = 0.002$ and $\rho = 7$.

We then test these schedules on the DDIM sampler Algorithm 1 by sampling images with $N = 10$ steps from the CIFAR-10 and CelebA models. We see that in Table 1 that our schedule improves the

---

[2]DDIM is usually presented using not $\sigma_t$ but parameters $\alpha_t$ satisfying $\sigma_t^2 = (1 - \alpha_t)/\alpha_t$. Linear updates of $\sigma_t$ are less natural when expressed in terms of $\alpha_t$.

| Sampler | CIFAR-10 FID | | | | CelebA FID | | | |
|---|---|---|---|---|---|---|---|---|
| | $N=5$ | $N=10$ | $N=20$ | $N=50$ | $N=5$ | $N=10$ | $N=20$ | $N=50$ |
| Ours | **12.53** | **3.85** | **3.39** | **3.43** | **10.73** | **4.30** | 3.56 | 3.78 |
| DDIM (Song et al., 2020a) | 47.20 | 16.86 | 8.28 | 4.81 | 32.21 | 18.08 | 11.81 | 7.39 |
| PNDM (Liu et al., 2022) | 13.9 | 7.03 | 5.00 | 3.95 | 11.3 | 7.71 | 5.51 | 3.34 |
| DPM (Lu et al., 2022a) | | 6.37 | 3.72 | **3.48** | | 5.83 | **2.82** | **2.71** |
| DEIS (Zhang & Chen, 2022) | 18.43 | 7.12 | 4.53 | 3.78 | 25.07 | 6.95 | 3.41 | 2.95 |
| UniPC (Zhao et al., 2023) | 23.22 | **3.87** | | | | | | |
| A-DDIM (Bao et al., 2022) | | 14.00 | 5.81* | 4.04 | | 15.62 | 9.22* | 6.13 |

Table 2: FID scores of our sampler compared to that of other samplers for pretrained CIFAR-10 and CelebA models with a discrete linear schedule. The first half of the table shows our computational results whereas the second half of the table show results taken from the respective papers. *Results for $N = 25$

FID of the DDIM sampler on both datasets even without the second-order updates. This is in part due to choosing a smaller $\sigma_N$ so the small number of steps can be better spent on lower noise levels (the difference between "DDIM" and "DDIM Offset"), and also because our schedule decreases $\sigma_t$ at a faster rate than DDIM (the difference between "DDIM Offset" and "Ours").

## 6.2 EVALUATION OF FULL SAMPLER

We quantitatively evaluate our sampler (Algorithm 2) by computing the Fréchet inception distance (FID) (Heusel et al., 2017) between all the training images and 50k generated images. We use denoisers from Ho et al. (2020); Song et al. (2020a) that were pretrained on the CIFAR-10 (32x32) and CelebA (64x64) datasets (Krizhevsky et al., 2009; Liu et al., 2015). We compare our results with other samplers using the same denoisers. The FID scores

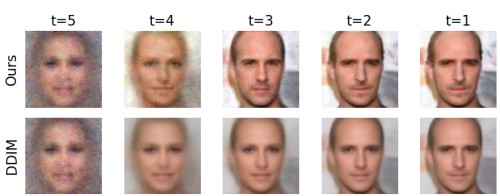

Figure 5: A comparison of our sampler with DDIM on the CelebA dataset with $N = 5$ steps.

are tabulated in Table 2, showing that our sampler achieves better performance on both CIFAR-10 (for $N = 5, 10, 20, 50$) and CelebA (for $N = 5, 10$).

We also incorporated our sampler into Stable Diffusion (a latent diffusion model). We change the noise schedule $\sigma_t$ as described in Appendix F. In Figure 1, we show some example results for text to image generation in $N = 10$ function evaluations, as well as FID results on 30k images generated from text captions drawn the MS COCO (Lin et al., 2014) validation set. From these experiments we can see that our sampler performs comparably to other commonly used samplers, but with the advantage of being much simpler to describe and implement.

## 7 RELATED WORK AND DISCUSSION

**Learning diffusion models** Diffusion models were originally introduced in Sohl-Dickstein et al. (2015) as using a variational inference method to learn the reverse of a process that progressively adds noise to data. This approach resulted in an improved training process (Ho et al., 2020; Nichol & Dhariwal, 2021) that becomes (1), which is different from the original variational lower bound. This improvement is justified from the perspective of denoising score matching (Song & Ermon, 2019; Song et al., 2020b), where the $\epsilon_\theta$ is interpreted as $\nabla \log(p(x_t, \sigma_t))$, the gradient of the log density of the data distribution perturbed by noise.

Score matching is also shown to be equivalent to denoising autoencoders with Gaussian noise (Vincent, 2011). From this derivation we can obtain a connection to our interpretation when $\mathcal{K}$ is a finite set of training examples. The *ideal denoiser* (Karras et al., 2022) for this setting is defined

as the minimizer of $\mathbf{E}_{x_0 \in \mathcal{K}} \mathbf{E}_{\epsilon \sim \mathcal{N}(0,1)} \| D(x_0 + \sigma\epsilon, \sigma) - x_0 \|^2$, which is equivalent to a smoothed version of the projection operator (6) of $\mathcal{K}$, with its argmin operation replaced by a "soft argmin" induced by the log-sum-exp function.

**Sampling from diffusion models** Samplers for diffusion models started with probabilistic methods (e.g. Ho et al. (2020)) that formed the reverse process by conditioning on the denoiser output at each step. In parallel, score based models (Song & Ermon, 2019; Song et al., 2020b) interpret the forward noising process as a stochastic differential equation (SDE), so SDE solvers based on Langevian dynamics (Welling & Teh, 2011) are employed to reverse this process. As models get larger, computational constraints motivated the development of more efficient samplers. Song et al. (2020a) then discovered that for smaller number of sampling steps, deterministic samplers perform better than stochastic ones. These deterministic samplers are constructed by reversing a non-Markovian process that leads to the same training objective, which is equivalent to turning the SDE into an ordinary differential equation (ODE) that matches its marginals at each sampling step.

This led to a large body of work focused on developing ODE and SDE solvers for fast sampling of diffusion models, a few of which we have evaluated in Table 2. Most notably, Karras et al. (2022) put existing samplers into a common framework and isolated components that can be independently improved. Our sampler Algorithm 2 bears most similarity to linear multistep methods, which can also be interpreted as accelerated gradient descent (Scieur et al., 2017). What differs is the error model: ODE solvers aim to minimize discretization error whereas we aim to minimize gradient estimation error, resulting in different "optimal" samplers.

**Linear-inverse problems and conditioning** Several authors (Kadkhodaie & Simoncelli, 2020; Chung et al., 2022; Kawar et al., 2022) have devised samplers for finding images that satisfy linear equations $Ax = b$. Such linear inverse problems generalize inpainting, colorization, and compressed sensing. In our framework, we can interpret this samplers as algorithms for equality constraint minimization of the distance function, a classical problem in optimization. Similarly, the widely used technique of *conditioning* (Dhariwal & Nichol, 2021) can be interpreted as multi-objective optimization, where minimization of distance is replaced with minimization of $\mathrm{dist}_{\mathcal{K}}(x)^2 + g(x)$ for an auxiliary objective function $g(x)$.

**Learning the distance function** Reinterpreting denoising as projection, or equivalently gradient descent on the distance function, has a few immediate implications. First, it suggests generalizations that draw upon the literature for computing distance functions and projection operators. Such techniques include Fast Marching Methods (Sethian, 1996), kd-trees, and neural-network approaches, e.g., Park et al. (2019); Rick Chang et al. (2017). Using concentration inequalities, we can also interpret training a denoiser as learning a solution to the *Eikonal PDE*, given by $\|\nabla d(x)\| = 1$. Other techniques for solving this PDE with deep neural nets include Smith et al. (2020); Lichtenstein et al. (2019); bin Waheed et al. (2021).

## 8 Conclusion, Limitations and Future Work

We have presented an elementary framework for analyzing and generalizing diffusion models that has led to a new sampling approach and new interpretations of pre-existing techniques. Moreover, the key objects in our analysis —the distance function and the projection operator—are canonical objects in constrained optimization. We believe our work can lead to new generative models that incorporate sophisticated objectives and constraints for a variety of applications. We also believe this work can be leveraged to incorporate existing denoisers into optimization algorithms in a plug-in-play fashion, much like the work in Chan et al. (2016); Le Pendu & Guillemot (2023); Rick Chang et al. (2017).

The limitations of our theory include its reliance on *reach*. While estimating reach is studied (Fefferman et al., 2016; Aamari et al., 2019), it is unclear if the reach of practically important datasets (e.g., the image manifold) can be estimated in practice. The correspondence between projection and denoising relies on the assumption that the manifold has low-dimension. If this assumption fails, the denoiser must be replaced with a different function that explicitly learns the projection operator. We think combining the multi-level noise paradigm of diffusion with distance function learning (Park et al., 2019) is an interesting direction, as are diffusion-models that carry out projection using analytic formulae or simple optimization routines.

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

## A  EQUIVALENT DEFINITIONS OF DDIM AND DDPM

The DDPM and DDIM samplers are usually described in a different coordinate system $z_t$ defined by parameters $\bar{\alpha}_t$ and the following relations , where the noise model is defined by a schedule $\bar{\alpha}_t$:

$$y \approx \sqrt{\bar{\alpha}_t}z + \sqrt{1 - \bar{\alpha}_t}\epsilon, \tag{13}$$

with the estimate $\hat{z}_0^t := \hat{z}_0(z_t, t)$ given by

$$\hat{z}_0(y, t) := \frac{1}{\sqrt{\bar{\alpha}_t}}(y - \sqrt{1 - \bar{\alpha}_t}\epsilon'_\theta(y, t)). \tag{14}$$

We have the following conversion identities between the $x$ and $z$ coordinates:

$$x_0 = z_0, \quad x_t = z_t/\sqrt{\bar{\alpha}_t}, \quad \sigma_t = \sqrt{\frac{1 - \bar{\alpha}_t}{\bar{\alpha}_t}}, \quad \epsilon_\theta(y, \sigma_t) = \epsilon'_\theta(y/\sqrt{\bar{\alpha}_t}, t). \tag{15}$$

While this change-of-coordinates is used in Song et al. (2020a, Section 4.3) and in Karras et al. (2022)–and hence not new– we rigorously prove equivalence of the DDIM and DDPM samplers given in Section 2 with their original definitions.

**DDPM**  Given initial $z_N$, the DDPM sampler constructs the sequence

$$z_{t-1} = \frac{\sqrt{\bar{\alpha}_{t-1}}(1 - \alpha_t)}{1 - \bar{\alpha}_t}\hat{z}_0^t + \frac{\sqrt{\alpha_t}(1 - \bar{\alpha}_{t-1})}{1 - \bar{\alpha}_t}z_t + \sqrt{\frac{1 - \bar{\alpha}_{t-1}}{1 - \bar{\alpha}_t}(1 - \alpha_t)}w_t, \tag{16}$$

where $\alpha_t := \bar{\alpha}_t/\bar{\alpha}_{t-1}$ and $w_t \sim \mathcal{N}(0, I)$. This is interpreted as sampling $z_{t-1}$ from a Gaussian distribution conditioned on $z_t$ and $\hat{z}_0^t$ (Ho et al., 2020).

**Proposition A.1** (DDPM change of coordinates). *The sampling update* (3) *is equivalent to the update* (16) *under the change of coordinates* (15).

*Proof.*  First we write (3) in terms of $z_t$, $\epsilon'_\theta(z_t, t)$ and $w_t$ using (14):

$$z_{t-1} = \frac{\sqrt{\bar{\alpha}_{t-1}}(1 - \alpha_t)}{\sqrt{\bar{\alpha}_t}(1 - \bar{\alpha}_t)}\left(z_t - \sqrt{1 - \bar{\alpha}_t}\epsilon'_\theta(z_t, t)\right) + \frac{\sqrt{\alpha_t}(1 - \bar{\alpha}_{t-1})}{1 - \bar{\alpha}_t}z_t + \sqrt{\frac{1 - \bar{\alpha}_{t-1}}{1 - \bar{\alpha}_t}(1 - \alpha_t)}w_t$$

$$= \frac{z_t}{\sqrt{\alpha_t}} + \frac{\alpha_t - 1}{\sqrt{\alpha_t(1 - \bar{\alpha}_t)}}\epsilon'_\theta(z_t, t) + \sqrt{\frac{1 - \bar{\alpha}_{t-1}}{1 - \bar{\alpha}_t}(1 - \alpha_t)}w_t.$$

Next we divide both sides by $\sqrt{\bar{\alpha}_{t-1}}$ and change $z_t$ and $z_{t-1}$ to $x_t$ and $x_{t-1}$:

$$x_{t-1} = x_t + \frac{\alpha_t - 1}{\sqrt{\bar{\alpha}_t(1 - \bar{\alpha}_t)}}\epsilon_\theta(x_t, \sigma_t) + \sqrt{\frac{1 - \bar{\alpha}_{t-1}}{\bar{\alpha}_{t-1}}\frac{1 - \alpha_t}{1 - \bar{\alpha}_t}}w_t.$$

Now if we define

$$\eta := \sqrt{\frac{1 - \bar{\alpha}_{t-1}}{\bar{\alpha}_{t-1}}\frac{1 - \alpha_t}{1 - \bar{\alpha}_t}} = \sigma_{t-1}\sqrt{\frac{1 - \bar{\alpha}_t/\bar{\alpha}_{t-1}}{1 - \bar{\alpha}_t}},$$

$$\sigma_{t'} := \sqrt{\sigma_{t-1}^2 - \eta^2} = \sigma_{t-1}\sqrt{\frac{\bar{\alpha}_t(1/\bar{\alpha}_{t-1} - 1)}{1 - \bar{\alpha}_t}} = \frac{\sigma_{t-1}^2}{\sigma_t},$$

it remains to check that

$$\sigma_{t'} - \sigma_t = \frac{\sigma_{t-1}^2 - \sigma_t^2}{\sigma_t} = \frac{1/\bar{\alpha}_{t-1} - 1/\bar{\alpha}_t}{\sqrt{1 - \bar{\alpha}_t}/\sqrt{\bar{\alpha}_t}} = \frac{\alpha_t - 1}{\sqrt{\bar{\alpha}_t(1 - \bar{\alpha}_t)}}.$$

$\square$

**DDIM**   Given initial $z_N$, the DDIM sampler constructs the sequence

$$z_{t-1} = \sqrt{\bar{\alpha}_{t-1}}\hat{z}_0^t + \sqrt{1 - \bar{\alpha}_{t-1}}\epsilon'_\theta(z_t, t), \tag{17}$$

i.e., it estimates $\hat{z}_0^t$ from $z_t$ and then constructs $z_{t-1}$ by simply updating $\bar{\alpha}_t$ to $\bar{\alpha}_{t-1}$. This sequence can be equivalently expressed in terms of $\hat{z}_0^t$ as

$$z_{t-1} = \sqrt{\bar{\alpha}_{t-1}}\hat{z}_0^t + \sqrt{\frac{1 - \bar{\alpha}_{t-1}}{1 - \bar{\alpha}_t}}(z_t - \sqrt{\bar{\alpha}_t}\hat{z}_0^t). \tag{18}$$

**Proposition A.2** (DDIM change of coordinates)**.** *The sampling update* (4) *is equivalent to the update* (18) *under the change of coordinates* (15)*.*

*Proof.* First we write (17) in terms of $z_t$ and $\epsilon'_\theta(z_t, t)$ using (14):

$$z_{t-1} = \sqrt{\frac{\bar{\alpha}_{t-1}}{\bar{\alpha}_t}}z_t + \left(\sqrt{1 - \bar{\alpha}_{t-1}} - \sqrt{\frac{\bar{\alpha}_{t-1}}{\bar{\alpha}_t}}\sqrt{1 - \bar{\alpha}_t}\right)\epsilon'_\theta(z_t, t).$$

Next we divide both sides by $\sqrt{\bar{\alpha}_{t-1}}$ and change $z_t$ and $z_{t-1}$ to $x_t$ and $x_{t-1}$:

$$\begin{aligned} x_{t-1} &= x_t + \left(\sqrt{\frac{1 - \bar{\alpha}_{t-1}}{\bar{\alpha}_{t-1}}} - \sqrt{\frac{\bar{\alpha}_{t-1}}{1 - \bar{\alpha}_t}}\right)\epsilon_\theta(x_t, \sigma_t) \\ &= x_t + (\sigma_{t-1} - \sigma_t)\epsilon_\theta(x_t, \sigma_t). \end{aligned}$$

$\square$

# B   FORMAL COMPARISON OF DENOISING AND PROJECTION

Our proof uses local Lipschitz continuity of the projection operator, stated formally as follows.

**Proposition B.1** (Theorem 6.2(vi), Chapter 6 of Delfour & Zolésio (2011))**.** *Suppose* $0 <$ $\text{reach}(\mathcal{K}) < \infty$. *Consider* $h > 0$ *and* $x, y \in \mathbb{R}^n$ *satisfying* $0 < h < \text{reach}(\mathcal{K})$ *and* $\text{dist}_\mathcal{K}(x) \leq h$ *and* $\text{dist}_\mathcal{K}(y) \leq h$. *Then the projection map satisfies* $\|\text{proj}_\mathcal{K}(y) - \text{proj}_\mathcal{K}(x)\| \leq \frac{\text{reach}(\mathcal{K})}{\text{reach}(\mathcal{K})-h}\|y - x\|$.

Decomposing random noise $\sigma\epsilon$ as

$$\sigma\epsilon = w_N + w_T \tag{19}$$

for $w_N \in N_\mathcal{K}(x_0)$ and $w_T \in N_\mathcal{K}(x_0)^\perp$ and using Lemma 3.1 allows us to show that $\text{proj}_\mathcal{K}(x_\sigma) \approx x_0$.

**Theorem B.1** (Denoising vs Projection)**.** *Fix* $\sigma > 0$ *and suppose* $\mathcal{K}$ *and* $t > 0$ *satisfies* $\text{reach}(\mathcal{K}) > \sigma(\sqrt{n} + t)$. *Given* $x_0 \in \mathcal{K}$ *and* $\epsilon \sim \mathcal{N}(0, I)$, *let* $x_\sigma = x_0 + \sigma\epsilon$ *and* $w := \sigma\epsilon = w_N + w_T$ *by the decomposition* (19)*. The following statements hold with probability at least* $1 - \exp(-\alpha t^2)$, *where* $\alpha > 0$ *is an absolute constant.*

- *(Backward error)* $x_0 = \text{proj}_\mathcal{K}(x_\sigma - w_T)$.

- *(Forward error)* $\|\text{proj}_\mathcal{K}(x_\sigma) - x_0\| \leq C\sigma(\sqrt{d} + t)$, *where* $C = \frac{\text{reach}(\mathcal{K})}{\text{reach}(\mathcal{K})-\sigma(\sqrt{n}+t)}$.

*Proof.* Let $B \in \mathbb{R}^{n \times d}$ denote an orthonormal basis for $N_\mathcal{K}(x_0)^\perp$, such that $w_T = BB^T w$, $\|w_T\| = \|B^T w\|$ and we have

$$\mathbf{E}[\|w_T\|^2] = \mathbf{E}[\|B^T w\|]^2 + \text{Tr}\,\text{cov}(B^T w) = \text{Tr}\,\text{cov}(B^T w) = \sigma^2\,\text{Tr}\,B^T B = \sigma^2 d. \tag{20}$$

Using a standard concentration inequality (Vershynin, 2018, page 44, Equation 3.3), we get that for a universal constant $\alpha$, with probability at least $1 - \exp(-\alpha t^2)$, we have $\|\epsilon\| \leq \sqrt{n} + t$ and $\|w_T\| \leq \sigma(\sqrt{d} + t)$. Using Lemma 3.1 and the fact that $\|w_N\| \leq \|\sigma\epsilon\| \leq \sigma(\sqrt{n} + t) < \text{reach}(\mathcal{K})$, we get

$$\text{proj}(x_\sigma - w_T) = \text{proj}(x_0 + w_N) = x_0,$$

proving the first statement. To prove the second statement, we observe that

$$
\begin{aligned}
\| \operatorname{proj}(x_\sigma) - x_0 \| &= \| \operatorname{proj}(x_0 + w_N + w_T) - x_0 \| \\
&= \| \operatorname{proj}(x_0 + w_N) - x_0 + \operatorname{proj}(x_0 + w_N + w_T) - \operatorname{proj}(x_0 + w_N) \| \\
&= \| \operatorname{proj}(x_0 + w_N) - \operatorname{proj}(x_0 + w_N + w_T) \| \\
&\leq C \| w_T \| \\
&\leq C \sigma(\sqrt{n} + t)
\end{aligned}
$$

where the second-to-last inequality comes from Proposition B.1, the assumption that $\operatorname{reach}(\mathcal{K}) > \sigma(\sqrt{n}+t)$, and the inequalities $\operatorname{dist}_\mathcal{K}(x_0 + w_N) \leq \| w_N \| \leq \sigma(\sqrt{n}+t)$ and $\operatorname{dist}_\mathcal{K}(x_0 + w_N + w_T) \leq \| w \| \leq \sigma(\sqrt{n}+t)$. □

## C  DDIM WITH PROJECTION ERROR ANALYSIS

### C.1  PROOF OF THEOREM 4.1

Make the inductive hypothesis that $\operatorname{dist}(x_t) = \sqrt{n}\sigma_t$. From the definition of DDIM (4), we have

$$
x_{t-1} = x_t + (\frac{\sigma_{t-1}}{\sigma_t} - 1)\sigma_t \epsilon_\theta(x_t, \sigma_t).
$$

Under Assumption 1 and the inductive hypothesis, we conclude

$$
\begin{aligned}
x_{t-1} &= x_t + (\frac{\sigma_{t-1}}{\sigma_t} - 1)\nabla f(x_t) \\
&= x_t - \beta_t \nabla f(x_t)
\end{aligned}
$$

Using Lemma 4.1 we have that

$$
\operatorname{dist}(x_{t-1}) = (1 - \beta_t)\operatorname{dist}(x_t) = \frac{\sigma_{t-1}}{\sigma_t}\operatorname{dist}(x_t) = \sqrt{n}\sigma_{t-1}
$$

The base case holds by assumption, proving the claim.

### C.2  PROOF OF LEMMA 4.1

Letting $x_0 = \operatorname{proj}_\mathcal{K}(x)$ and noting $\nabla f(x) = x - x_0$, we have

$$
\begin{aligned}
\operatorname{dist}_\mathcal{K}(x_+) &= \operatorname{dist}_\mathcal{K}(x + \beta(x_0 - x)) \\
&= \| x + \beta(x_0 - x) - x_0 \| \\
&= \| (x - x_0)(1 - \beta) \| \\
&= (1 - \beta)\operatorname{dist}_\mathcal{K}(x)
\end{aligned}
$$

### C.3  PROOF OF LEMMA 4.2

By (Delfour & Zolésio, 2011, Chapter 6, Theorem 2.1), $|\operatorname{dist}_\mathcal{K}(u) - \operatorname{dist}_\mathcal{K}(v)| \leq \| u - v \|$, which is equivalent to

$$
\operatorname{dist}_\mathcal{K}(u) - \operatorname{dist}_\mathcal{K}(v) \leq \| u - v \|, \operatorname{dist}_\mathcal{K}(v) - \operatorname{dist}_\mathcal{K}(u) \leq \| u - v \|.
$$

Rearranging proves the claim.

### C.4  PROOF OF LEMMA 4.3

We first restate the full version of Lemma 4.3.

**Lemma C.1.** *For $\mathcal{K} \subseteq \mathbb{R}^n$, let $f(x) := \frac{1}{2}\operatorname{dist}_\mathcal{K}(x)^2$. The following statements hold.*

  (a) *If $x_+ = x - \beta(\nabla f(x) + e)$ for $e$ satisfying $\| e \| \leq \eta \operatorname{dist}_\mathcal{K}(x)$ and $0 \leq \beta \leq 1$, then*

$$
(1 - \beta(\eta + 1))\operatorname{dist}_\mathcal{K}(x) \leq \operatorname{dist}_\mathcal{K}(x_+) \leq (1 + \beta(\eta - 1))\operatorname{dist}_\mathcal{K}(x).
$$

*(b) If $x_{t-1} = x_t - \beta_t(\nabla f(x_t) + e_t)$ for $e_t$ satisfying $\|e_t\| \leq \eta \mathrm{dist}_\mathcal{K}(x_t)$ and $0 \leq \beta_t \leq 1$, then*

$$\mathrm{dist}_\mathcal{K}(x_N) \prod_{i=t}^{N}(1 - \beta_i(\eta + 1)) \leq \mathrm{dist}_\mathcal{K}(x_{t-1}) \leq \mathrm{dist}_\mathcal{K}(x_N) \prod_{i=t}^{N}(1 + \beta_i(\eta - 1).)$$

For Item (a) we apply Lemma 4.2 at points $u = x_+$ and $v = x - \beta \nabla f(x)$. We also use $\mathrm{dist}(v) = (1 - \beta)\mathrm{dist}_\mathcal{K}(x)$, since $0 \leq \beta \leq 1$, to conclude that

$$(1 - \beta)\mathrm{dist}_\mathcal{K}(x) - \beta\|e\| \leq \mathrm{dist}_\mathcal{K}(x_+) \leq (1 - \beta)\mathrm{dist}_\mathcal{K}(x) + \beta\|e\|.$$

Using the assumption that $\|e\| \leq \eta \mathrm{dist}_\mathcal{K}(x)$ gives

$$(1 - \beta - \eta\beta)\mathrm{dist}_\mathcal{K}(x) \leq \mathrm{dist}_\mathcal{K}(x_+) \leq (1 - \beta + \eta\beta)\mathrm{dist}_\mathcal{K}(x)$$

Simplifying completes the proof. Item (b) follows from Item (a) and induction.

### C.5 PROOF OF THEOREM 4.2

We first state and prove an auxillary theorem:

**Theorem C.1.** *Suppose Assumption 2 holds for $\nu \geq 1$ and $\eta > 0$. Given $x_N$ and $\{\beta_t, \sigma_t\}_{i=1}^{N}$, recursively define $x_{t-1} = x_t + \beta_t \sigma_t \epsilon_\theta(x_t, t)$ and suppose that $\mathrm{proj}_\mathcal{K}(x_t)$ is a singleton for all $t$. Finally, suppose that $\{\beta_t, \sigma_t\}_{i=1}^{N}$ satisfies $\frac{1}{\nu}\mathrm{dist}_\mathcal{K}(x_N) \leq \sqrt{n}\sigma_N \leq \nu\mathrm{dist}_\mathcal{K}(x_N)$ and*

$$\frac{1}{\nu}\mathrm{dist}_\mathcal{K}(x_N) \prod_{i=t}^{N}(1 + \beta_i(\eta - 1)) \leq \sqrt{n}\sigma_{t-1} \leq \nu\mathrm{dist}_\mathcal{K}(x_N) \prod_{i=t}^{N}(1 - \beta_i(\eta + 1)). \quad (21)$$

*The following statements hold.*

- $\mathrm{dist}_\mathcal{K}(x_N) \prod_{i=t}^{N}(1 - \beta_i(\eta + 1)) \leq \mathrm{dist}_\mathcal{K}(x_{t-1}) \leq \mathrm{dist}_\mathcal{K}(x_N) \prod_{i=t}^{N}(1 + \beta_i(\eta - 1))$

- $\frac{1}{\nu}\mathrm{dist}_\mathcal{K}(x_{t-1}) \leq \sqrt{n}\sigma_{t-1} \leq \nu\mathrm{dist}_\mathcal{K}(x_{t-1})$

*Proof.* Since $\mathrm{proj}_\mathcal{K}(x_t)$ is a singleton, $\nabla f(x_t)$ exists. Hence, the result will follow from (7) in Lemma 4.3 if we can show that $\|\beta_t \sigma_t \epsilon_\theta(x_t, t) - \nabla f(x_t)\| \leq \eta \mathrm{dist}_\mathcal{K}(x_t)$. Under Assumption 2, it suffices to show that

$$\frac{1}{\nu}\mathrm{dist}_\mathcal{K}(x_t) \leq \sqrt{n}\sigma_t \leq \nu\mathrm{dist}_\mathcal{K}(x_t) \quad (22)$$

holds for all $t$. We use induction, noting that the base case ($t = N$) holds by assumption. Suppose then that (22) holds for all $t, t + 1, \ldots, N$. By Lemma 4.3 and Assumption 2, we have

$$\mathrm{dist}_\mathcal{K}(x_N) \prod_{i=t}^{N}(1 - \beta_i(\eta + 1)) \leq \mathrm{dist}_\mathcal{K}(x_{t-1}) \leq \mathrm{dist}_\mathcal{K}(x_N) \prod_{i=t}^{N}(1 + (\eta - 1)\beta_i)$$

Combined with (21) shows

$$\frac{1}{\nu}\mathrm{dist}_\mathcal{K}(x_{t-1}) \leq \sqrt{n}\sigma_{t-1} \leq \nu\mathrm{dist}_\mathcal{K}(x_{t-1}),$$

proving the claim. □

The proof of Theorem 4.2 follows that of Theorem C.1 by additionally observing $\eta < 1$ implies that $\mathrm{dist}_\mathcal{K}(x_t) < \mathrm{reach}(\mathcal{K})$ for all $t$, which implies $\mathrm{proj}_\mathcal{K}(x_t)$ is a singleton.

### C.6 PROOF OF THEOREM 4.3

Assuming constant step-size $\beta_i = \beta$ and dividing (8) by $\prod_{i=1}^{N}(1 - \beta)$ gives the conditions

$$\left(1 + \eta\frac{\beta}{1 - \beta}\right)^N \leq \nu, \qquad \left(1 - \eta\frac{\beta}{1 - \beta}\right)^N \geq \frac{1}{\nu}.$$

Rearranging and defining $a = \eta\frac{\beta}{1-\beta}$ and $b = \nu^{\frac{1}{N}}$ gives

$$a \leq b - 1, \qquad a \leq 1 - b^{-1}.$$

Since $b - 1 - (1 - b^{-1}) = b + b^{-1} - 2 \geq 0$ for all $b > 0$, we conclude $a \leq b - 1$ holds if $a \leq 1 - b^{-1}$ holds. We therefore consider the second inequality $\eta\frac{\beta}{1-\beta} \leq 1 - \nu^{-1/N}$, noting that it holds for all $0 \leq \beta < 1$ if and only if $0 \leq \beta \leq \frac{k}{1+k}$ for $k = \frac{1}{\eta}(1 - \nu^{-1/N})$, proving the claim.

## C.7 Proof of Theorem 4.4

The value of $\sigma_0/\sigma_N$ follows from the definition of $\sigma_t$ and and the upper bound for $\mathrm{dist}_\mathcal{K}(x_0)/\mathrm{dist}_\mathcal{K}(x_N)$ follows from Theorem 4.3. We introduce the parameter $\mu$ to get a general form of the expression inside the limit:

$$(1 - \mu\beta_{*,N})^N = \left(1 - \mu\frac{1 - \nu^{-1/N}}{\eta + 1 - \nu^{-1/N}}\right)^N.$$

Next we take the limit using L'Hôpital's rule:

$$
\begin{aligned}
\lim_{N\to\infty} \left(1 - \mu\frac{1 - \nu^{-1/N}}{\eta + 1 - \nu^{-1/N}}\right)^N &= \exp\left(\lim_{N\to\infty} \log\left(1 - \mu\frac{1 - \nu^{-1/N}}{\eta + 1 - \nu^{-1/N}}\right)/(1/N)\right) \\
&= \exp\left(\lim_{N\to\infty} \frac{\eta\mu\log(\nu)}{(\nu^{-1/N} - \eta - 1)(\nu^{1/N}(\eta - \mu + 1) + \mu - 1)}\right) \\
&= \exp\left(-\frac{\mu\log(\nu)}{\eta}\right) \\
&= (1/\nu)^{\mu/\eta}.
\end{aligned}
$$

For the first limit, we set $\mu = 1$ to get

$$\lim_{N\to\infty} (1 - \beta_{*,N})^N = (1/\nu)^{1/\eta}.$$

For the second limit, we set $\mu = 1 - \eta$ to get

$$\lim_{N\to\infty} (1 + (\eta - 1)\beta_{*,N})^N = (1/\nu)^{\frac{1-\eta}{\eta}}.$$

## C.8 Denoiser Error

Assumption 2 places a condition directly on the approximation of $\nabla f(x)$, where $f(x) := \frac{1}{2}\mathrm{dist}_\mathcal{K}(x)$, that is jointly obtained from $\sigma_t$ and the denoiser $\epsilon_\theta$. We prove this assumption holds under a direct assumption on $\nabla\mathrm{dist}_\mathcal{K}(x)$, which is easier to verify in practice.

**Assumption 3.** *There exists $\nu \geq 1$ and $\eta > 0$ such that if $\frac{1}{\nu}\mathrm{dist}_\mathcal{K}(x) \leq \sqrt{n}\sigma_t \leq \nu\mathrm{dist}_\mathcal{K}(x)$ then $\|\epsilon_\theta(x,t) - \sqrt{n}\nabla\mathrm{dist}_\mathcal{K}(x)\| \leq \eta$*

**Lemma C.2.** *If Assumption 3 holds with $(\nu, \eta)$, then Assumption 2 holds with $(\hat{\nu}, \hat{\eta})$, where $\hat{\eta} = \frac{1}{\sqrt{n}}\eta\nu + \max(\nu - 1, 1 - \frac{1}{\nu})$ and $\hat{\nu} = \nu$.*

*Proof.* Multiplying the error-bound on $\epsilon_\theta$ by $\sigma_t$ and using $\sqrt{n}\sigma_t \leq \nu\mathrm{dist}_\mathcal{K}(x)$ gives

$$\|\sigma_t\epsilon_\theta(x,t) - \sqrt{n}\sigma_t\nabla\mathrm{dist}_\mathcal{K}(x)\| \leq \eta\sigma_t \leq \eta\nu\frac{1}{\sqrt{n}}\mathrm{dist}_\mathcal{K}(x)$$

Defining $C = \sqrt{n}\sigma_t - \mathrm{dist}_\mathcal{K}(x)$ and simplifying gives

$$
\begin{aligned}
\eta\nu\frac{1}{\sqrt{n}}\mathrm{dist}_\mathcal{K}(x) &\geq \|\sigma_t\epsilon_\theta(x,t) - \sqrt{n}\sigma_t\nabla\mathrm{dist}_\mathcal{K}(x)\| \\
&= \|\sigma_t\epsilon_\theta(x,t) - \nabla f(x) - C\nabla\mathrm{dist}_\mathcal{K}(x)\| \\
&\geq \|\sigma_t\epsilon_\theta(x,t) - \nabla f(x)\| - \|C\nabla\mathrm{dist}_\mathcal{K}(x)\| \\
&= \|\sigma_t\epsilon_\theta(x,t) - \nabla f(x)\| - |C|
\end{aligned}
$$

Since $(\frac{1}{\nu} - 1)\mathrm{dist}_\mathcal{K}(x) \leq C \leq (\nu - 1)\mathrm{dist}_\mathcal{K}(x)$ and $\nu \geq 1$, the Assumption 2 error bound holds for the claimed $\hat{\eta}$. $\qquad\square$

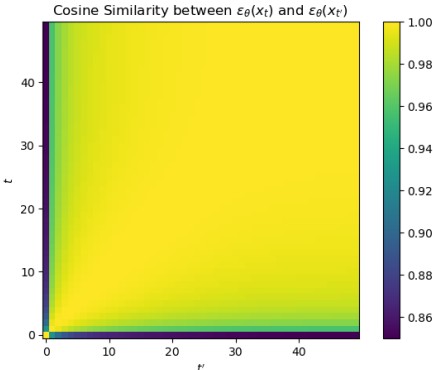

Figure 6: Plot of the cosine similarity between $\epsilon_\theta(x_t, t)$ and $\epsilon_\theta(x_{t'}, t')$ over $N = 50$ steps of DDIM denoising on the CIFAR-10 dataset. Each cell is the average result of 1000 runs.

## D  DERIVATION OF GRADIENT ESTIMATION SAMPLER

To choose $W$, we make two assumptions on the denoising error: the coordinates $e_t(\epsilon)_i$ and $e_t(\epsilon)_j$ are uncorrelated for all $i \neq j$, and $e_t(\epsilon)_i$ is only correlated with $e_{t+1}(\epsilon)_i$ for all $i$. In other words, we consider $W$ of the form

$$W = \begin{bmatrix} aI & bI \\ bI & cI \end{bmatrix} \tag{23}$$

and next show that this choice leads to a simple rule for selecting $\bar{\epsilon}$. From the optimality conditions of the quadratic optimization problem (11), we get that

$$\bar{\epsilon}_t = \frac{a+b}{a+c+2b}\epsilon_\theta(x_t, \sigma_t) + \frac{c+b}{a+c+2b}\epsilon_\theta(x_{t+1}, \sigma_{t+1}).$$

Setting $\gamma = \frac{a+b}{a+c+2b}$, we get the update rule (12). When $b \geq 0$, the minimizer $\bar{\epsilon}_t$ is a simple convex combination of denoiser outputs. When $b < 0$, we can have $\gamma < 0$ or $\gamma > 1$, i.e., the weights in (12) can be negative (but still sum to 1). Negativity of the weights can be interpreted as cancelling positively correlated error ($b < 0$) in the denoiser outputs. Also note we can implicitly search over $W$ by directly searching for $\gamma$.

## E  FURTHER EXPERIMENTS

### E.1  DENOISING APPROXIMATES PROJECTION

We test our interpretation that denoising approximates projection on pretrained diffusion models on the CIFAR-10 dataset. In these experiments, we take a 50-step DDIM sampling trajectory, extract $\epsilon(x_t, \sigma_t)$ for each $t$ and compute the cosine similarity for every pair of $t, t' \in [1, 50]$. The results are plotted in Figure 6. They show that the direction of $\epsilon(x_t, \sigma_t)$ over the entire sampling trajectory is close to the first step's output $\epsilon(x_N, \sigma_N)$. On average over 1000 trajectories, the minimum similarity (typically between the first step when $t = 50$ and last step when $t' = 1$) is 0.85, and for the vast majority (over 80%) of pairs the similarity is $> 0.99$, showing that the denoiser outputs approximately align in the same direction, validating our intuitive picture in Figure 2.

### E.2  DISTANCE FUNCTION PROPERTIES

We test Assumption 1 and Assumption 2 on pretrained networks. If Assumption 1 is true, then $\|\epsilon_\theta(x_t, \sigma_t)\| \sqrt{n} = \|\nabla \mathrm{dist}_\mathcal{K}(x_t)\| = 1$ for every $x_t$ along the DDIM trajectory. In Figure 7a, we plot the distribution of norm of the denoiser $\epsilon_\theta(x_t, \sigma_t)$ over the course of many runs of the DDIM sampler on the CIFAR-10 model for $N = 100$ steps ($t = 1000, 990, \ldots, 20, 10, 0$). This plot shows that $\|\epsilon_\theta(x_t, \sigma_t)\| / \sqrt{n}$ stays approximately constant and is close to 1 until the end of the sampling

(a) Plot of $\|\epsilon_\theta(x_t, \sigma_t)\| / \sqrt{n}$ against $t$.      (b) Plot of $\|\epsilon_\theta(x_0 + \sigma_t\epsilon, \sigma_t) - \epsilon\| / \sqrt{n}$ against $t$.

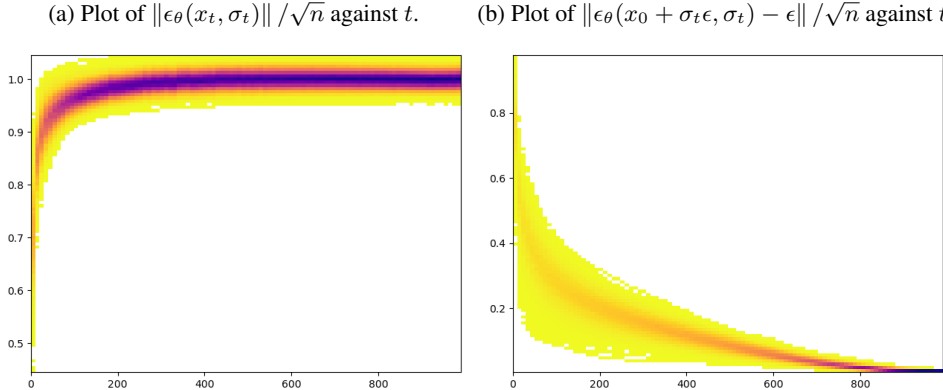

Figure 7: Plots of the norm of the denoiser at different stages of denoising, as well as the ability of the denoiser to accurately predict the added noise as a function of noise added.

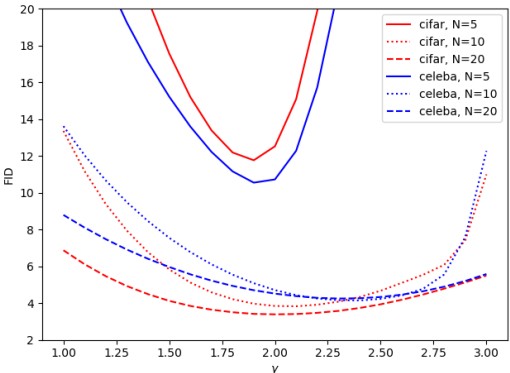

Figure 8: Plot of FID score against $\gamma$ for our second-order sampling algorithm on the CIFAR-10 and CelebA datasets for $N = 5, 10, 20$ steps.

process. We next test Assumption 3, which implies Assumption 2 by Lemma C.2. We do this by first sampling a fixed noise vector $\epsilon$, next adding different levels of noise $\sigma_t$, then using the denoiser to predict $\epsilon_\theta(x_0 + \sigma_t\epsilon, \sigma_t)$. In Figure 7b, we plot the distribution of $\|\epsilon_\theta(x_0 + \sigma_t\epsilon, \sigma_t) - \epsilon\| / \sqrt{n}$ over different levels of $t$, as a measure of how well the denoiser predicts the added noise.

### E.3 CHOICE OF $\gamma$

We motivate our choice of $\gamma = 2$ in Algorithm 2 with the following experiment. For varying $\gamma$, Figure 8 reports FID scores of our sampler on the CIFAR-10 and CelebA models for $N = 5, 10, 20$ timesteps using the $\sigma_t$ schedule described in Appendix F.3. As shown, $\gamma \approx 2$ achieves the optimal FID score over different datasets and choices of $N$.

## F EXPERIMENT DETAILS

### F.1 PRETRAINED MODELS

The CIFAR-10 model and architecture were based on that in Ho et al. (2020), and the CelebA model and architecture were based on that in Song et al. (2020a). The specific checkpoints we use are provided by Liu et al. (2022). We also use Stable Diffusion 2.1 provided in `https://huggingface.co/stabilityai/stable-diffusion-2-1`. For the comparison experiments in Figure 1, we implemented our gradient estimation sampler to interface with the Hug-

gingFace diffusers library and use the corresponding implementations of UniPC, DPM++, PNDM and DDIM samplers with default parameters.

## F.2 FID SCORE CALCULATION

For the CIFAR-10 and CelebA experiments, we generate 50000 images using our sampler and calculate the FID score using the library in `https://github.com/mseitzer/pytorch-fid`. The statistics on the training dataset were obtained from the files provided by Liu et al. (2022). For the MS-COCO experiments, we generated images from 30k text captions drawn from the validation set, and computed FID with respect to the 30k corresponding images.

## F.3 OUR SELECTION OF $\sigma_t$

Let $\sigma_1^{\text{DDIM}(N)}$ be the noise level at $t = 1$ for the DDIM sampler with $N$ steps. For the CIFAR-10 and CelebA models, we choose $\sigma_1 = \sqrt{\sigma_1^{\text{DDIM}(N)}}$ and $\sigma_0 = 0.01$. For CIFAR-10 $N = 5, 10, 20, 50$ and CelebA $N = 5$ we choose $\sigma_N = 40$ and for CelebA $N = 10, 20, 50$ we choose $\sigma_N = 80$. For Stable Diffusion, we use the same sigma schedule as that in DDIM.

## F.4 TEXT PROMPTS

For the text to image generation in Figure 1, the text prompts used are:

- "A digital Illustration of the Babel tower, 4k, detailed, trending in artstation, fantasy vivid colors"
- "London luxurious interior living-room, light walls"
- "Cluttered house in the woods, anime, oil painting, high resolution, cottagecore, ghibli inspired, 4k"

