# OpenReview forum: "Interpreting and improving diffusion models using the Euclidean distance function"
_ICLR.cc/2024/Conference — Submitted to ICLR 2024_

### Official Review · Reviewer_LpGF · 2023-10-31

**Soundness:** 2 fair
**Presentation:** 2 fair
**Contribution:** 1 poor
**Rating:** 3
**Confidence:** 4

**Summary:**

This paper is focused on presenting an interpretation of diffusion models, in which each iteration interpreted as a non-linear projection onto the data manifold. They define a Euclidian distance function from noisy image to the manifold, and interpret the denoising step as the gradient descent on this distance function. On the empirical side, the paper proposes a modified update line based on the theory, with the goal of minimizing and correcting the score prediction error. When number of iterations are limited to small numbers, this method achieved better performance in FID compared to other methods, on CIFAR10 and celebA.

**Strengths:**

The writing is clear and the math is reasonably easy to follow.
The derivations are generally sound and well written.

**Weaknesses:**

The main weakness of this work is that theorem 3.1 rests on a strong assumption that is violated in the case of image manifolds. Up to this theorem, the math is not novel and is borrowed from other sources. The contribution of this work is in Theorem 3.1, where the existing math about non-linear projection is used to interpret backward diffusion/denoising. This theorem is the foundation for the rest of the paper, and the derivations about the bounds sit on top of this theorem. The problem is that the theorem rests on a strong assumption on the reach of the image manifold: $reach(K) \geq \sigma \sqrt{n}$ which does not necessarily hold. This assumes the curvature of the manifold is small compared to $\sigma$. This greatly simplifies the geometry of the assumed manifold but the problem is such manifold is too simplistic to approximate *image* manifold. If images lie on manifold, those manifolds must be complex with high curvatures.

To make it clear as to why image manifolds must contain high curvature (or small reach) it is enough to run a simple mental experiment. Consider an image, which represents a single point on the manifold. All the invariances of that image, such as local translations, also exist within this manifold. The geometry of local translations is well understood in the Fourier domain. When an image is translated, its Fourier amplitude remains constant, while its Fourier phases vary. Consequently, translated versions of an image lie on circles, with the circle's radius equivalent to the Fourier amplitude. It's a widely known empirical observation that the power spectrum of images adheres to a $1/f$ law, where $f$ represents the frequency, stating that higher frequencies exhibit smaller amplitudes. Therefore, the circles on the manifold, due to the translation invariance of image details (higher frequencies), have small radii. The $1/f$ phenomenon is an empirical but firmly established characteristic of images.

The reach(K) of image manifold is not a constant and depends on the image (i.e. the curvature varies across the manifold). But as long as the image is not blank and contains details, the $reach(K)$ will be small. For a typical image, the reach can be as small as the quantization precision of the images in the dataset, due to the $1/f$ property: the highest frequencies create small circles around a given image.

The analysis in the paper does not hold unless for very small sigma (in the vicinity of the manifold), where we can assume the sigma is smaller than the radius of the curvature. But the paper claims this analysis can be used for denoising in diffusion models which always start with very large noise, and the noise stays large throughout the intermediate steps until towards the very end of the trajectory.

Moreover, the claim that the gradient doesn't change does not hold if the assumption on reach is violated. In other words, when the curvature is larger than sigma, the gradient does change direction at each step (up to a point where sigma is small enough where the manifold is locally flat relative to sigma level).

**Questions:**

Please see weaknesses section for comments.

---

> ### Author Response · Authors · 2023-11-16
>
> We thank the reviewer for the critical review. The main objection raised in this review is the dependence on reach in Theorem 3.1. The reviewer claims that the reach assumption is unrealistic for image manifolds.
>
> Our main theoretical contribution is Theorem 4.2, which depends only on the existence of a unique projection and Assumption 2, which states that the denoiser is an approximate projection in a relative sense. In Sections 3 and 4 we outlined the reasons for why we believe this is a reasonable assumption, using Theorem 3.1 when the noise level is small and arguing that for large noise levels both the denoising and projection vectors approximately point in the same direction.
>
> Note that Assumption 2 can be tested empirically (see the experiments in Appendix E), and Theorem 4.2 does not require any condition on the reach of the manifold. Theorem 3.1, which requires a condition on reach, is only used to motivate the definition of Assumption 2.

---

### Official Review · Reviewer_dfzc · 2023-11-01

**Soundness:** 3 good
**Presentation:** 4 excellent
**Contribution:** 3 good
**Rating:** 6
**Confidence:** 3

**Summary:**

The authors reinterpret denoising diffusion models as approximate gradient descent applied to the Euclidean distance function under manifold hypothesis. In the interpretation, they establish rigorous connection between denoising function and the projection to data manifold, which is equivalent to the gradient of squared distance function to the manifold. They also reframe the convergence analysis of diffusion models using the new interpretation, providing a justification for the well-known log-linear noise schedule used in diffusion models.

Building on these insights, the authors introduce a higher-order sampler of diffusion models. This sampler leverages the invariant properties of projection (projection is invariant along line segments between a point and the its projection), regularizing that the direction at one point in one step is similar to the direction at the point of next step. Their new sampler does not require additional evaluation of denoising (score) function and achieves SOTA FID scores on pretrained CIFAR-10 and CelebA models.

**Strengths:**

This paper offers a novel interpretation of DDIM samplers, framing them as approximate gradient descent applied to the squared distance from data manifold. The convergence analysis of DDIM with Euclidean distance function instead of divergence over probability space adds to its novelty.

The authors propose improved schedules and samplers for DDIM based on their theoretical findings, which enhances the practical applicability of proposed theory. This improvement upon previous models based on theoretical foundations is notable.

**Weaknesses:**

The proposed schedules and samplers are tested on a limited number of small-scale datasets

Implications of lemma/theorem are unclear
- Thm4.2: In rea settings, is final sample error (dist_K(x_0)) bounded?
- Thm4.3: Why does beta star suddenly appear?
- Thm4.4: Can we adopt beta for achieving convergence to zero? And how does it related to the schedule showcased in Section 6
- Section5: What are the meaning of a positive definite metric and the implications of the metric with gamma equal to 2?

**Questions:**

It would be beneficial for the authors to relate their work to existing research regarding to geometry of diffusion models, such as "Score-Based Generative Models Detect Manifolds," "A Geometric Perspective on Diffusion Models," and "Understanding the Latent Space of Diffusion Models through the Lens of Riemannian Geometry

---

> ### Author Response · Authors · 2023-11-16
>
> We thank the reviewer for the helpful comments and references. Regarding the comment on only testing the schedules and samplers on small-scale datasets, we also conducted experiments on high-resolution latent-diffusion models such as Stable diffusion. The text-to-image FID scores on 30k samples using the MS-COCO dataset are stated in Fig. 1, with more details in Section 6.2.
>
> Below we provide some clarifications on the implications of the theorems:
>
> * Thm 4.2: Under our relative-error Assumption 2, Theorem 4.2 characterizes the noise schedules that would enable convergence of the DDIM algorithm, as measured by the distance to the data manifold. Intuitively, if the noise schedule has too large steps, the premise of Assumption 2 would be violated during the sampling process and convergence may not be guaranteed. This motivates our definition of admissible schedules and the theorem justifies the need for multiple steps of denoising in diffusion models.
> Regarding bounds of $\text{dist}(x_0)$ in practice, this is possible when the error parameters ($\nu$, $\eta$) can be computed.  For this, one can use ground truth projections (if available). It may also be possible to bound ($\nu$, $\eta$) using denoiser training error, but we defer this to future research.
> * Thm 4.3: We apologize if the theorem as stated is unclear.  $\beta^*$ is defined in the statement of Theorem 4.3.  As the theorem establishes, it equals the largest admissible step-size for constant sigma schedules.  Equivalently, it defines the fastest log-linear decrease of sigma.
> * Thm 4.4: This is an excellent question. Unfortunately, we must defer its answer to future research given the short rebuttal period.
> * Section 5: This weighting matrix can be interpreted as the inverse covariance matrix of positively correlated denoiser error. See appendix D.

---

> > ### Comment · Reviewer_dfzc · 2023-11-21
> >
> > Thank you for addressing my questions. However, I still believe the proof of concept in this paper falls short when compared to concurrently accepted papers. Moreover, considering the concerns raised by reviewer LpGF, it is essential to highlight a critical theoretical flaw in the paper. The mathematical concept of “reach” remains unexplored in the context of data manifolds and requires thorough investigation before being employed to derive new implications about diffusion models. Given these reservations, I have developed skepticism regarding to the acceptance of this work at ICLR.

---

> > > ### Author Response · Authors · 2023-11-21
> > >
> > > Thank you for the response. We would like to reiterate that the reach assumption only affects the results in Sections 2 and 3, and our main results and analysis in Section 4 do not depend on the reach assumption. They only depend on the projection with relative error assumption (Assumption 2), which is motivated by our discussion in Section 3.  This is clarified In our response to reviewer LpGF.
> > >
> > > In summary, our main results in Section 4 depend on Assumption 2, which states that the neural net $\epsilon_\theta(x,t)$ returns an approximate projection when its input $x$ has approximately $\sigma_t$ level of noise. This is motivated by the fact that $\epsilon_\theta(x,t)$  is trained on inputs $x = x_0 + \epsilon \sigma_t$, and the discussion at the end of Section 3, which as we illustrate in Figure 2b can hold whenever the noise is small compared to reach or _when the noise is large compared to the diameter of the data set_. We believe this is a reasonable assumption and it is tested experimentally on real image datasets in Appendix E.
> > >
> > > Informally, Theorem 4.2 characterizes what $\sigma_t$ schedule we can use to guarantee convergence when minimizing $\text{dist}_\mathcal{K}(x_t)$. In particular, it tells us that we cannot take too few steps or too large steps.
> > > - At large $\sigma_t$, taking a large step will have bounded _relative error_ but may have large _absolute error_, thus we cannot denoise with a single step.
> > > - At small $\sigma_t$, the bounded _relative error_ implies a small _absolute error_, but the input to $\epsilon_\theta(x,t)$ must have small levels of noise to begin with.
> > > - Thus we need to iteratively reduce $\sigma_t$ in a controlled manner leading to the definition of _admissible schedules_
> > >
> > > Theorem 4.2 also directly informs the log-linearly decreasing $\sigma_t$ schedule we use when improving the sampling process (Algorithm 2).

---

### Official Review · Reviewer_gUw5 · 2023-11-07

**Soundness:** 3 good
**Presentation:** 2 fair
**Contribution:** 3 good
**Rating:** 8
**Confidence:** 3

**Summary:**

In this paper, the authors offer a novel perspective by interpreting denoising diffusion models as an approximation of gradient descent applied to the Euclidean distance function. Furthermore, the paper includes a comprehensive convergence analysis of DDIM. The experimental results underscore the effectiveness of the proposed sampler, as it attains state-of-the-art FID scores and consistently generates high-quality samples.

**Strengths:**

This paper presents a compelling blend of theoretical interpretation and practical enhancements. While the projection interpretation of diffusion models is not entirely novel, the authors provide a more concrete example by framing sampling as an approximation of gradient descent on the distance function to the training dataset.

**Weaknesses:**

.

**Questions:**

Q1.The results presented in Table 2 appear to be incomplete, with certain experiments seemingly omitted. It would be beneficial to provide a more comprehensive set of results to ensure a thorough evaluation.

Q2. Additionally, the comparison with DPM suggests that DPM performs comparably to the proposed sampler. It might be helpful to provide further insights or analysis regarding the similarities and differences between the two methods to clarify their relative strengths and weaknesses

---

> ### Author Response · Authors · 2023-11-16
>
> We thank the reviewer for the feedback. As for the questions raised:
> 1. The results in the second half of the table are taken from the respective papers and thus some values are missing because they are not originally reported.
> 2. Compared to the DPM sampler, our sampler performs better when using fewer sampling steps (i.e. using $N=5,10$ function evaluations), and also has the advantage of being much simpler to describe and implement.

---

### Author Response · Authors · 2023-11-21

We thank the reviewers again for their comments and we have responded to them in the replies below. Please let us know if there are any additional questions and we are happy to answer them in the remainder of the discussion period.

---

### Meta-Review · Area_Chair_qRnM · 2023-12-14

**Metareview:**

The paper studies deterministic diffusion solvers through the lens of manifold projections. Diffusion models are often motivated through a manifold hypothesis — as well structured ways of generating samples from a manifold of valid images. The paper notes that manifold denoising is closely related to manifold projection, in the sense that when the noise is small relative to the reach of the manifold (small enough that projections “stay local”), projecting a noisy point onto the manifold recovers the clean signal up to an error of sigma \sqrt{d}, where d is the intrinsic dimension. The paper’s main novelties are, first, to analyze the convergence of deterministic diffusion solvers, under the assumption that the denoising step of these solvers accurately approximates a projection, in the sense that \sigma_t \eps(x,t) is a relative error approximation of the gradient of the squared distance function. This allows the paper to analyze a perturbed gradient descent on the squared distance function, which exhibits linear convergence. This analysis yields constraints on step sizes and smoothing levels for convergence. The second novelty is to propose a momentum-like method, which the paper motivates by noting that for the projection problem, the gradient direction stays constant (since gradient descent always moves in the normal direction to the manifold). Experiments show improved performance (FID scores) of the momentum-like solver compared to a baseline deterministic solver (DDIM).

**Justification For Why Not Higher Score:**

The paper generated mixed reviews.

On the positive side, the paper contributes to the recent study of connections between diffusion models and data geometry, and provides a new perspective on the choice of smoothing schedules and step sizes. In experiment, the proposed momentum / averaging method improves over the baseline solver. Relating relatively complicated diffusion models to the simpler geometric operation of projection makes it easier to study the effect of geometry on these models behavior, making this a promising line of inquiry.


One reviewer concern focuses on the conditions of the paper’s theoretical claims — namely, the paper’s analysis assumes that the denoiser is an approximate projection (in the sense of Assumptions 1 and 2). As Section 3 of the paper notes, projections are only unique locally; the largest euclidean neighborhood of the manifold on which the paper’s analysis holds is dictated by the reach of the manifold. For certain image manifolds, the reach is small. Intuitively, this is due to the presence of sharp edges — e.g., it continuum image articulation manifolds are not even differentiable submanifolds of image space (e.g., Wakin et al The Multiscale Structure of Non-Differentiable Image Manifolds). Because the few image manifolds that can be studied analytically have large curvature (at least before smoothing / regularization), it is a commonly held intuition that image manifolds corresponding to high resolution / sharp images are highly curved, and hence have small very reach.

At the same time, the reach is a worst case quantity (over all points on the manifold, and all normal directions). There may be many points and directions along which the projection is unique, even beyond this relatively restrictive scale. As noted by the authors, this could mitigate the small reach concern, since the observed behavior of diffusion samplers is generated by typical samples from a gaussian distribution, which may not be worst-case, in the sense of the reach.


The paper uses Assumptions 1 and 2 as an interface between denoising and projection: as both the authors and reviewers note, Section 3 is important background (and key intuition for why denoising is related to projection), but not a strong novelty of the paper. The main contributions are in Section 4-5, which, through Assumption 2, assumes that the denoiser approximates the gradient of the squared distance function, and then works out convergence results for perturbed gradient descent on this function. In this regard, the mathematics does not completely substantiate the main message of the paper, which, in the AC's reading is that denoisers in diffusion models can be treated as approximate projections onto the data manifold -- at a technical level, the paper assumes this.

The main message of the paper would be better substantiated (and the work would be much stronger as theory), if Assumption 2 were proved, rather than asserted. If Assumption 2 could be shown for “ideal” denoisers eps_*(x, sigma) (e.g., MMSE denoiser under population measure, or the score function) which neural net denoisers are trained to approximate, this would be a highly significant result, as it would make rigorous the central contention of the paper — that denoising diffusion models can be interpreted as computing projections onto the data manifold. In the AC’s assessment, this would be highly interesting even if the result was restricted to “below reach” scale. Assumption 2 does not follow in a clear way from the statement of Theorem 3.1, since this bounds the forward and backward error between proj_K(x) and x_0, rather than between proj_K(x) and eps_*(x, sigma).

In the AC's reading, at least some of the questions around the role of the reach come because the paper does not make clear what the precise, rigorous relationship between Theorem 3.1 and Assumption 2 is. If proofs of Assumption 2 are not possible, devoting more of the paper to justifying this assumption experimentally would also improve the paper. In current form, given its mixed evaluation, the paper falls below the bar for acceptance.

**Justification For Why Not Lower Score:**

N/A

---

### Decision · Program_Chairs · 2024-01-16

Reject